# Diagnostic Utility of Cytomegalovirus (CMV) DNA Quantitation in Ulcerative Colitis

**DOI:** 10.3390/v16050691

**Published:** 2024-04-26

**Authors:** Sema Esen, Imran Saglik, Enver Dolar, Selcan Cesur, Nesrin Ugras, Harun Agca, Osman Merdan, Beyza Ener

**Affiliations:** 1Department of Medical Microbiology, Bursa Uludag University Hospital, Bursa 16120, Turkeyharunagca@uludag.edu.tr (H.A.); osmanmerdan@uludag.edu.tr (O.M.); bener@uludag.edu.tr (B.E.); 2Department of Gastroenterology, Bursa Uludag University Hospital, Bursa 16120, Turkey; edolar@uludag.edu.tr (E.D.); selcancesur@uludag.edu.tr (S.C.); 3Department of Medical Pathology, Bursa Uludag University Hospital, Bursa 16120, Turkey; nesrinugras@uludag.edu.tr

**Keywords:** cytomegalovirus, ulcerative colitis, cytomegalovirus colitis, quantitative PCR, CMV DNA, antiviral treatment

## Abstract

Cytomegalovirus (CMV) colitis is a critical condition associated with severe complications in ulcerative colitis (UC). This study aimed to investigate the diagnostic value of the presence of CMV DNA in intestinal mucosa tissue and blood samples in patients with active UC. This study included 81 patients with exacerbated symptoms of UC. Patient data were obtained from the Hospital Information Management System. CMV DNA in colorectal tissue and plasma samples were analyzed using a real-time quantitative PCR assay. CMV markers were detected using immunohistochemistry and hematoxylin–eosin staining. Immunohistochemistry positivity was observed in tissue samples from eight (9.9%) patients. Only one (1.2%) patient showed CMV-specific intranuclear inclusion bodies. CMV DNA was detected in 63.0% of the tissues (median: 113 copies/mg) and in 58.5% of the plasma samples (median: 102 copies/mL). For tissues, sensitivity and the negative predictive value (NPV) for qPCR were excellent (100.0%), whereas specificity and the positive predictive value (PPV) were low (41.9% and 15.7%, respectively). For plasma, sensitivity and NPV were high (100.0%) for qPCR, whereas specificity and PPV were low (48.6% and 24.0%, respectively). CMV DNA ≥392 copies/mg in tissue samples (sensitivity 100.0% and specificity 83.6%) and ≥578 copies/mL (895 IU/mL) in plasma samples (sensitivity 66.7% and specificity 100.0%) provided an optimal diagnosis for this test. The qPCR method improved patient management through the early detection of CMV colitis in patients with UC. However, reliance on qPCR positivity alone can lead to overdiagnosis. Quantification of CMV DNA can improve diagnostic specificity, although standardization is warranted.

## 1. Introduction

Cytomegalovirus (CMV) is an opportunistic pathogen that can cause primary infection or secondary reactivations. After primary infection, CMV establishes lifelong persistence in various organs and tissues of the body. Reactivations of CMV can lead to severe clinical manifestations in affected organs in immunosuppressed individuals [1,2,3]. The risk of reactivation is higher in the intestinal tissues of patients with inflammatory bowel disease (IBD), specifically, ulcerative colitis (UC). The colon is a common site affected by CMV reactivation in the gastrointestinal system (GIS), resulting in acute CMV colitis [3,4,5]. According to Kwon et al., the likelihood of CMV colitis is 19- and 31-fold higher in patients with IBD and UC, respectively [4]. The prevalence of CMV-associated colitis ranges from 10% to 17% in patients with severe IBD [5]. Patients with corticosteroid-resistant UC are at an increased risk of developing CMV colitis [2,3,4].

CMV can reactivate as a result of the effects of UC and immunosuppressive treatments, such as malnutrition and dysfunction of host defense factors [2,3,4,5,6]. Simultaneously, inflammation and mucosal damage in the bowel mucosa leads to the secretion of proinflammatory cytokines, which stimulate CMV-infected macrophage migration to the damaged mucosal tissue. Progressive inflammation and vasculitis cause severe ischemia and transmural necrosis. In addition, increasing mucosal erosion and deep ulcers damage blood vessels and cause bloody diarrhea [3,6]. In contrast, CMV has proinflammatory effects, promoting the expression of cytokines, chemokines, and cellular adhesion molecules that may activate the primary disease.

The symptoms of CMV colitis resembling the exacerbation period of the primary disease (UC) include abdominal pain, diarrhea, rectal bleeding, and fever. In progressive cases, the infection may lead to complications, such as fulminant colitis, toxic megacolon, or perforation [3,4,6].

Colonoscopy is recommended in appropriate cases, and biopsies of mucosal lesions should be used for the specific and differential diagnosis of CMV colitis. Diffuse edematous mucosa, patchy erythema, exudates, multiple mucosal erosions, deep ulcers, and pseudopolyps may be revealed on colonoscopic examination [7,8]. 

Several diagnostic approaches can be used for suspected intestinal CMV infection. Detecting viral markers through histological examination is essential for diagnosing tissue-invasive CMV infection. Hematoxylin–eosin (H&E) staining is a conventional method that can reveal typical intracellular viral inclusions (owl-eye appearance) with high specificity but low sensitivity. The detection of CMV antigens on tissue by immunohistochemistry (IHC) provides more sensitive evidence of active infection and is considered the gold standard [2,3,5,9,10]. However, IHC is not standardized and is less sensitive than CMV DNA detection by quantitative real-time PCR (qPCR) [2,3,11]. The PCR method, widely used in clinical practice currently with increasing applicability, measures viral load in a few hours with the availability of automation [5,10,11,12].

CMV can be detected in various samples, including blood, intestinal tissue biopsy, and stool. However, interpreting the test results is difficult, and whether the detection of CMV DNA in patient samples is an incidental reactivation due to immunosuppression or a trigger for an attack is controversial [9,11]. No consensus has been reached regarding the relationship between viral load and CMV colitis [5,12]. Differences have been reported in the diagnosis and management of CMV colitis. When CMV is detected in intestinal tissue (with PCR or IHC), there is a common tendency (82%) among gastroenterologists to initiate antiviral therapy [12]. This study aimed to investigate the diagnostic value of the presence of CMV DNA in colorectal mucosal biopsy tissue and blood samples in patients with active UC.

## 2. Materials and Methods

### 2.1. Study Population

This prospective study included 81 patients who underwent colonoscopy for UC exacerbation at the Bursa Uludag University Health Application and Research Center between January 2019 and March 2022. Patients were excluded from this study if bacteria such as *Clostridium difficile*, *Salmonella* spp., or parasites, including Entamoeba histolytica, were detected by culture, serology, or direct microscopy of stool samples. All patients included in this study were CMV-seropositive.

Patient data obtained from electronic records in the Hospital Information Management System included patient demographics; GIS symptoms such as abdominal pain, stool frequency, diarrhea, or bloody diarrhea; mucosal appearance at endoscopy; treatment; and outcome. Patients with UC were evaluated for severity using the Mayo score and categorized as mild, moderate, or severe. To diagnose CMV colitis, patients had to meet the following specific criteria: exhibit clinical symptoms consistent with colitis, such as diarrhea or bloody stools, and undergo pathological testing to confirm CMV tissue involvement.

### 2.2. Colonoscopy

A gastroenterologist performed a colonoscopy to evaluate the mucosa. Biopsies were taken from inflamed areas and the edges of ulcers in the mucosa using sterile endoscopic biopsy forceps. Four to five tissue samples, weighing 10–37 mg each, were obtained from various locations. Two of the biopsy specimens were promptly placed in 1 mL of 0.9% saline and transferred to the microbiology department for qPCR analysis. Simultaneously, the other two samples were fixed in 10% buffered formalin and sent to the Pathology Department.

### 2.3. Preparation of Tissue Samples for QPCR

The sample weight was measured under sterile conditions. The samples were then mixed with 1000 μL of lysis buffer (Abbott, Chicago, IL, USA) and gently shaken in a water bath at 45 °C for 6 h to homogenize them [13]. The following steps were carried out in a similar manner to those for the plasma samples. Finally, the CMV load per milligram of tissue was determined based on the qPCR results.

### 2.4. qPCR CMV DNA Assay

CMV DNA was analyzed using the Abbott RealTime CMV assay (Abbott, Chicago, IL, USA) from colorectal mucosa tissue of all patients at the Microbiology Laboratory of Bursa Uludag University Health Application and Research Centre. Additionally, plasma samples from 43 patients were also tested using the same assay. Viral nucleic acid extraction and amplification were performed using a fully automated Abbott m2000 Molecular Analyzer (Abbott, Chicago, IL, USA). The limit of quantitation for CMV DNA was <20 copies/mL (c/mL). To determine the IU/mL value in plasma, the kit manufacturer recommends multiplying c/mL values by 1.55.

### 2.5. Immunohistochemical Examination

The Pathology Department evaluated the presence of inflammation and typical large cells with basophilic intranuclear CMV inclusions (cytomegalic cells) in colorectal mucosal tissue samples using H&E staining. Endothelial, epithelial, and stromal cells in mucosal tissue samples were examined with CMV (8B1.2, 1G5.2, and 2D4.2) mouse monoclonal antibodies (Cell Marque, Sigma-Aldrich, Mountain View, CA, USA) to detect CMV-specific proteins through IHC. Positive results were determined by typical CMV-specific nuclear or cytoplasmic staining.

### 2.6. Statistical Analysis

Data normality was assessed using the Shapiro–Wilk test. Two nonparametric tests (Kruskal–Wallis and Mann–Whitney U) were used to compare groups for nonnormally distributed data. Descriptive statistics are reported as median (minimum–maximum). Statistical comparisons of categorical data were performed using Pearson’s chi-squared test, Fisher’s exact test, and the Fisher–Freeman–Halton test. Descriptive statistics of categorical data are reported as frequencies and percentages. Relationships between variables were analyzed using Spearman’s rank correlation coefficient. We compared the sensitivity (SN) and specificity (SP) of qPCR and H&E staining analyses with those of IHC, the gold standard method accepted in this study. Optimal diagnostic viral load cutoffs for tissue and plasma samples were determined using Receiver Operating Characteristic (ROC) analysis with the Youden J index based on IHC assessment. This analysis allows for the determination of the SN and SP of a test, as well as the optimal cutoff for diagnosis. Data analysis was performed using SPSS v25 and MEDCALC 19.5.6 statistical software.

## 3. Results

This study included 81 adult patients with UC, with a median age of 45 years (range: 20–79). Factors affecting CMV DNA in tissue samples and risk factors associated with CMV colitis are presented in Table 1. Age and viral load exhibited a significant correlation, with higher levels of CMV DNA more prevalent in patients over 50 years of age (r = 0.401; *p* < 0.001). Additionally, there was a significant correlation between plasma viral load and age (r = 0.591; *p* < 0.001).

This study involved patients with active UC symptoms. The differences in tissue CMV DNA positivity and viral load levels between patients with diarrhea and patients with bloody diarrhea were found to be insignificant. The patients were categorized based on UC activity as follows [14]: mild (6.2%), moderate (24.7%), and severe (69.1%). However, differences in detecting or quantifying CMV DNA in tissues according to UC activity were also insignificant. IHC positivity was only observed in patients with moderate (10.0%, *n* = 2/20) and severe (10.7%, *n* = 6/56) disease (Table 1).

During the colonoscopy, intestinal biopsies were taken from the mucosal areas of the colon (*n* = 40) and rectum (*n* = 31) that exhibited hyperemia, edema, and exudate, indicating inflammation. This study evaluated the association between mucosal features and CMV in patients. Patients with intestinal mucosa showed a cobblestone-like appearance and pseudopolyps of 24% and 22%, respectively. Multiple mucosal ulcerations and erosions were observed in 86.4% of the patients. Ulcer characteristics, including aphthous ulcers (*n* = 7), geographic ulcers (*n* = 18), and deep ulcers (*n* = 11), with mucopurulent exudate, were available for only 36 patients. However, the correlation between the presence of ulcers and the detection of CMV DNA or viral load in tissue samples was insignificant (Table 1).

Similar to the results of colonoscopy, H&E staining revealed inflammation in the mucosal biopsies of all patients. The majority of patients (93.8%) exhibited chronic active inflammation, while a small percentage (6.2%) showed areas of chronic inflammation in the mucosa. The presence of CMV DNA was higher in patients with chronic active inflammation (65.8%) compared with those with chronic inflammation (20.0%). Additionally, CMV load levels were increased in patients with chronic active inflammation compared with patients with chronic inflammation (*p* = 0.040) (Table 1).

A significant percentage of the study group received medical treatment for UC, as shown in Table 2. Among the patients, seven (87.5%) of those who were IHC-positive were receiving corticosteroids. CMV DNA was found at a higher rate in the tissues of patients using mesalamine, corticosteroids, and azathioprine compared with those not using these agents. However, these differences were deemed insignificant (Table 2). Notably, levels of viral load and IHC positivity were higher in patients receiving corticosteroids (*p* = 0.059 and 0.007, respectively). No association was found between anti-infliximab and anti-adalimumab (monoclonal antibodies against TNF-alpha) and CMV infection.

In total, 21 patients with severe colitis and elevated CMV DNA levels (median: 20,151 c/mg; range: 392–160,159) in tissue samples were diagnosed with CMV colitis and administered ganciclovir/valganciclovir (2 × 5 mg/kg for 21 days). Test results for all patients, including those diagnosed with CMV colitis, are presented in Table 3 and Table 4. Among these patients, 13 were corticosteroid-refractory, 8 were positive for both IHC and tissue CMV-DNA, and 13 were only positive for CMV-DNA in tissue. According to clinical and laboratory data including plasma CMV DNA levels, remission was achieved in 90.5% (*n* = 19) of the patients who were followed up on an outpatient basis one month later (Table 4). CMV DNA was found in bronchoalveolar lavage samples from two patients with pulmonary involvement. A colectomy was performed on two patients who did not respond to treatment. In one patient, CMV DNA levels in the plasma continued to rise during follow-up, ultimately resulting in death. CMV reactivation, clinical progression, or complications were absent in patients (*n* = 60; 30 of whom were positive for CMV DNA, with a median of 20 copies/mg [c/mg], range: 20–287) who had not received antiviral treatment. Our findings suggest that if all CMV DNA-positive patients had been treated, 58.8% (30/51) of them would have received unnecessary antiviral drugs. 

Colorectal mucosa samples were evaluated through histopathological examination. Only one (1.2%) patient showed CMV-specific intranuclear inclusion bodies. CMV antigen positivity was observed in the tissue samples of eight (9.9%) patients upon histopathological examination using IHC (Figure 1). CMV DNA was found in 63.0% of the colorectal tissues of patients (median: 113 c/mg, range 0–160,159). It was also detected in 58.1% of the plasma samples (median: 102 c/mL, range 20–1937). CMV DNA was detected in tissues from all patients who were positive with IHC (*n* = 8) and H&E staining (*n* = 1). Patients who tested positive for IHC were also positive for both H&E staining and qPCR in both tissue and plasma samples. CMV DNA was detected by qPCR in plasma samples from 25 out of 43 patients, of whom 22 (88.0%) were also positive by qPCR in tissue samples.

Viral loads were higher (median 25,613.0 c/mg, range 477–160,159) in tissues of patients who were IHC-positive than IHC-negative (20.0 c/mg, range 0–16,182) (*p* < 0.001) (Table 1). Similarly, CMV DNA levels were higher (922.0 c/mL, range: 20–1937) in the plasma samples of patients who tested positive using the IHC method than in those who tested negative (20.0 c/mL [range: 0–578]) (*p* = 0.006, Table 5).

Compared with the qPCR results for plasma, CMV DNA was detected in a large proportion (88.0%, *n* = 22/25) of the plasma-positive tissue samples. Viral loads were higher in the tissues of patients with CMV DNA detected in the plasma than in patients without CMV DNA (500.0 c/mg [0–160,159] and 0.0 c/mg [0–510] (*p* = 0.001). The viral load levels in tissue and plasma samples showed a positive correlation (r = 0.741; *p* = 0.001). The plasma qPCR assay showed SN and SP values of 78.5% and 80.0%, respectively, compared with tissue samples. In comparison with IHC, the qPCR method showed excellent SN (100.0%) for tissue and plasma samples but lower SP (41.9% and 48.6%, respectively) and positive predictive value (PPV) (15.7% and 24.0%, respectively). By contrast, the qPCR method had an increased negative predictive value (NPV) for the detection of CMV in intestinal tissue (90.12%) and plasma samples (100.0%).

The results of this study indicate that qPCR analysis is the most sensitive method to detect CMV in tissues (AUC: 0.943 [95% CI: 0.87–0.88]; *p* < 0.001), followed by plasma (AUC: 0.842 [95% CI: 0.70–0.93]; *p* < 0.001) (Figure 2). The difference between tissue and blood in ROC analysis was not significant (*p* = 0.458). qPCR analysis of two tissue samples from different mucosal areas in 20 patients showed different results for 3 (15%) patients (Table 6).

## 4. Discussion

In active UC, it is crucial to differentiate between exacerbating the primary disease, CMV colitis, and clinically unrelated mild CMV reactivation, especially when managing immunomodulatory or antiviral treatment [3,4,8,15]. Studies have reported an incidence of CMV colitis in 4.5% of newly diagnosed cases of UC and 13.8% to 40% of severe steroid-refractory cases using IHC-based methods [6,10,15,16,17]. Approximately 10% of patients with UC had CMV colitis [3,4,5,6,17]. In this study, the CMV positivity rate was 9.9% with IHC in patients with active UC, consistent with the results of other studies.

The analysis of viral infections in different age groups showed a correlation between aging and the exacerbating role of the immune system in disease. These factors are also important for CMV colitis [1,18,19,20,21]. In this study group, age and viral load were significantly correlated (r = 0.401; *p* < 0.001), with higher positivity and levels of CMV DNA being particularly prevalent in patients over 50 years of age (*p* = 0.002 and *p* < 0.001, respectively). In addition, higher levels of CMV DNA (77.5 c/mg, range 0–160,159) were observed in the tissue samples of patients with previous CMV colitis (*p* = 0.014). The findings indicate a strong association between CMV reactivation and immune system function, which tends to decrease with age.

The clinical features of CMV colitis are well-known and similar to those of exacerbations in UC [22,23,24]. CMV colitis should be considered in cases of moderate-to-severe UC, and its prevalence in inactive or mild UC is insignificant [6,23,24]. According to Ko et al., CMV colitis is associated with symptoms such as hematochezia (51.0%), diarrhea (45.1%), fever (15.7%), abdominal pain (15.7%), nausea and vomiting (5.9%), and melena (7.8%) [22]. Our assessment of patient specifications showed that the majority of patients (93.8%) had moderate and severe UC activity. IHC positivity was only found in patients with moderate (10.0%, *n* = 2/20) and severe (10.7%, 6/56) UC activity. Similarly, tissue CMV DNA positivity was higher in patients with severe (64.3%, 36/56) and moderate (65.0%, 13/20) than mild (40.0%, 2/5) UC activity. However, the association between CMV DNA and disease activity and symptoms was insignificant. Although qPCR tests are the most sensitive for detecting CMV nucleic acids, they may not indicate tissue-invasive diseases that affect clinical outcomes. In particular, the detection of low levels of viral DNA does not indicate the formation of active viral particles within the cell or cause an acute pathological process. In addition, CMV infection in ulcerative colitis patients may be more related to long-term outcomes rather than current clinical presentations. However, this result may be due to the small number of patients with mild symptoms in this study.

Tissue-invasive CMV infection presents histopathologically with inflammation and ulceration with ulcers reported in 52–60% of patients [25,26,27]. Studies have associated positive CMV IHC results with severe CMV colitis, which may feature punched-out ulcers, irregular ulcers, or a cobblestone appearance [6,21,26]. Some reports imply that ulceration is a sign of disease severity rather than CMV infection [6,27]. In our study group, a high rate (86.4%) of mucosal ulcers was observed, and all patients who were IHC positive (*n* = 8) had ulceration. However, we did not find a significant correlation between ulcers and the detection or level of CMV DNA (*p* = 0.960 and 0.599). This statistical result may be due to CMV being a latent virus, or the small number of mild patients may have influenced this outcome. Including patients with varying degrees of UC severity in future studies could offer more insight into this field.

In this study, inflammation (chronic active inflammation or chronic inflammation) was detected in the mucosa of the colon and rectum of all patients. Patients with chronic active inflammation in H&E staining had a higher rate of CMV DNA positivity (65.8%) and a higher median viral load (20.0 c/mg, range 0–160,159) compared with those with chronic inflammation (CMV DNA positivity 20.0% and median viral load 0.0 c/mg [0–20]). The role of CMV infection in exacerbating the severity of inflammation in UC remains controversial [3,8]. Our data suggest that CMV reactivation and viral load are higher in active inflammation. Even if the level of CMV DNA is low, it can trigger an increased immune response and inflammation. However, silent infections that are not directly responsible for the current clinical situation may not be innocent, as they can lead to sustained inflammatory reactions and potential long-term effects. There may be a connection between the virus-mediated increase in inflammation during viral replication and the subsequent rise in viral replication caused by inflammation. Given the complex interactions between CMV and the immune system, it is crucial to explore the role of host genetics in the pathogenesis of CMV colitis.

Immunosuppression facilitates CMV reactivation. In recent decades, there has been a growing concern about opportunistic infections associated with the increased use of immunomodulators in patients with UC. Steroids can reduce T-cell function, leading to increased CMV replication. Therefore, CMV colitis is common in patients undergoing steroid treatment [2,3,4,19,21,28]. Hirayama et al. and others have reported that treatment with agents other than corticosteroids is not significantly associated with CMV colitis [20,21]. As expected, we found no association between monoclonal antibodies targeting TNF-alpha and CMV infection. However, the use of corticosteroids may indicate the severity of UC [29,30,31]. There is a significant risk of CMV infection, particularly with prolonged (>1 month) use of corticosteroids in combination with other immunosuppressive drugs [1,29,32]. In this study, CMV positivity by IHC was remarkably high (*p* = 0.007) in patients receiving steroids, consistent with previous reports. CMV DNA levels increased in the steroid-receiving group. The trigger for the first CMV reactivation may be related to inflammation of the intestinal tissue rather than the use of corticosteroids. However, corticosteroids may exacerbate clinically innocent CMV reactivation, and inflammation may become more severe and widespread because of an increased viral load and virus–immune relationships. CMV-seropositive societies, particularly those in developing nations, should be aware of this [33]. The incidence of CMV seropositivity is high in our country, and all of the patients in this study were CMV IgG positive.

CMV colitis is diagnosed by confirming the presence of CMV in tissues with symptoms and signs attributable to CMV. Identifying CMV through histopathological examination with H&E staining and IHC yields a precise diagnosis (92–100%) for active CMV infection in the intestine [2,5]. Inclusions caused by intracellular CMV particles are not always typical or obvious; H&E staining has the lowest SN (10–87%) [2,5,10]. In our study, inclusion bodies were seen in only one patient with H&E staining, and SN was 12.5%. The European Crohn’s and Colitis Organization recommends PCR for CMV DNA detection and IHC in intestinal tissue for CMV colitis [29]. The IHC method uses monoclonal antibodies to identify CMV antigens, improving diagnostic yield with increased SN compared with H&E staining. IHC evaluation offers the advantage of detecting active viral replication in tissues [5]. qPCR has the best diagnostic yield for CMV DNA detection in various samples, is objective and automated, and is useful for monitoring treatment response [3,9,34]. Detecting mild CMV reactivation in tissue unrelated to the clinic may lead to overdiagnosis and administration of antivirals [5,11]. CMV DNA quantification can be enlightening in this aspect. Yoshino et al. detected CMV DNA in 56.7% of colonic tissues, but only 5.9% tested positive with IHC [35]. According to reports, SN and SP of CMV qPCR on tissue range from 65.0% to 100.0% and 40.0% to 100.0%, respectively [2,5]. This study detected CMV DNA in 63.0% of the tissue samples but only 9.9% with IHC. In tissue samples, IHC, SN, and NPV were excellent (100.0%) for qPCR, whereas SP and PPV were low (41.9% and 15.7%, respectively). Cohen et al. found that the level of CMV DNA significantly correlated with the number of inclusion bodies on biopsy specimens with IHC [36]. In our study, patients with positive results in IHC had higher CMV DNA levels. Increased CMV DNA levels may be evidence of a tissue-invasive disease. These results explain the difference in SN performance between the two methods.

Blood samples may have lower concentrations of CMV than tissues in localized infections [5,15,37]. In cases of CMV colitis, the infection may remain limited to the colon, or viral nucleic acids may be detected in the blood a few days or weeks after CMV replication in the intestinal tissue because the virus can spread into the blood in later stages [6,29,34]. Compared with qPCR in tissues, the SN and SP values of qPCR in plasma samples were 78.5% and 80.0%, respectively. We detected CMV DNA in the plasma samples of all patients who tested positive for IHC in their tissues (*n* = 8). In addition, we found higher CMV loads (922.0 c/mL [20–1937]) in the plasma samples of patients who were positive than in those who were negative for IHC (20.0 c/mL [0–578]). Patients positive for tissue CMV DNA (n = 25) had higher median plasma viral load (86.5 c/mL [0–1937]) than patients negative for it (0.0 c/mL [0–105]). The correlation between viral loads measured in both samples was insignificant (r = 0.741; *p* < 0.001). ROC analysis indicated that qPCR was reliable for detecting CMV in tissues (AUC = 0.943) and plasma (AUC = 0.842), revealing similarities between tissue and blood samples (*p* = 0.458). Studies have reported that CMV qPCR testing in plasma has an SN of 65–100% and an SP of 40–92% [2,5]. In this study, compared with IHC, the SN and NPV of qPCR in plasma samples were excellent (100.0%), whereas SP and PPV were low (48.6% and 24.0%, respectively). These results suggest that examining CMV DNA in the plasma provides important insights into tissue CMV load. Quantifying the CMV load in plasma may be a viable alternative to colonoscopy for diagnosing patients, but further research is needed to validate its accuracy. However, in three patients, low levels of viremia (CMV DNA: 20, 44, and 105 c/mL) were detected only in plasma samples. Although CMV reactivation was considered in these patients, GIS involvement was not proven. Therefore, although detecting CMV DNA in blood samples is helpful in diagnosing CMV infection, it may not be evidence of GIS involvement.

This study has shown that detecting CMV DNA in tissue is highly sensitive for diagnosing CMV colitis. However, establishing diagnostic thresholds may be useful because of the low SP of qPCR. Algorithm efforts have been sustained to manage patients with UC and concomitant CMV infection [5]. Some studies have proposed a diagnostic cutoff level for CMV DNA [8]. For example, based on the antiviral treatment response, Roblin et al. suggested a CMV DNA level of 250 c/mg as a diagnostic threshold (SN 100%, SP 66%) [15]. Their study has been widely accepted as a reference by many authors and centers for a long time [5]. Mavropoulou et al. reported the highest SN (79%) and SP for qPCR results with >250 c/mg tissues in 47 IBD cases, while Paul et al. proposed ≥316 c/mg in 92 UC and nine Crohn’s disease cases [35,36]. Recently, Cohen et al. found a CMV DNA cutoff of 259 IU/mL for blood samples (Roche Cobas 6800), with an SN of 77% and an SP of 99%, to diagnose CMV colitis [37]. In this study, the optimal cutoff values (for the Abbot CMV DNA kit) provided for qPCR CMV DNA were ≥392 c/mg (SN 100.0% and SP 83.6%) of tissue and ≥578 c/mL (895 IU/mL) of plasma (SN 66.7% and SP 100.0%). According to this cutoff, the prevalence of CMV colitis in the study group was 25.9% (*n* = 21/81). To our knowledge, this study is one of the few that has evaluated CMV DNA levels to diagnose CMV colitis and proposed a cutoff value. However, qPCR CMV testing faces significant challenges, including standardization of quantitative data across laboratories and harmonization of viral load in tissue samples [36]. In addition, the PCR assays may exhibit quantitative variations.

Our study suggests that the detection of CMV DNA and its levels may not be directly linked to acute clinical symptoms. Notably, CMV infection may be a sign of immune dysfunction and increased inflammation, potentially serving as an early marker for severe colitis and complications. This is consistent with the findings of other studies [36,38].

CMV infection presents a patchy distribution in the intestines, even in individuals with severe disease (25). Roblin et al. found that CMV DNA was only present in inflamed mucosa, not in normal tissue from the same patient (15). The location and number of intestinal biopsies are essential for diagnosing CMV infection, and taking multiple colonic tissue biopsies is recommended to enhance SN (15,25). In this study, biopsies were taken from inflamed or ulcerated areas of the mucosa, with a high rate of CMV DNA detection (63.0%). Furthermore, it was observed that 3 out of 20 patients showed variations in CMV DNA between tissues. One sample from a patient was negative for CMV DNA, while another was positive, and the two patients had different viral loads. This finding suggests that analyzing only one tissue sample for diagnosing CMV colitis may lead to false negatives. It also indicates that analyzing multiple tissue samples improves SN. As a result of this study, our center has started reporting CMV analysis using both IHC and qPCR, with the evaluation of at least two tissue samples.

This study has limitations due to the small number of cases with mild clinical symptoms and the inability to obtain plasma CMV DNA from all patients. Patients were evaluated based on their clinical remissions and plasma CMV DNA levels. Using endoscopic responses to assess antiviral treatment may be more appropriate in future studies.

## 5. Conclusions

In conclusion, qPCR is very useful as an initial screening test for the early diagnosis of CMV colitis. It could improve patient management by facilitating the early detection of CMV colitis in patients with UC. However, overdiagnosis resulting from the qPCR method may lead to the unnecessary use of antivirals, causing undesirable side effects. In this study, a diagnosis based on CMV DNA positivity alone would have led to unnecessary antiviral treatment in about half of our patients (58.8%). The presence of steroid-resistant symptoms, blood CMV DNA positivity, and IHC positivity increases the specificity of tissue CMV DNA positivity. Particularly, quantification of CMV DNA may enhance diagnostic SP, but standardization is warranted.

## Figures and Tables

**Figure 1 viruses-16-00691-f001:**
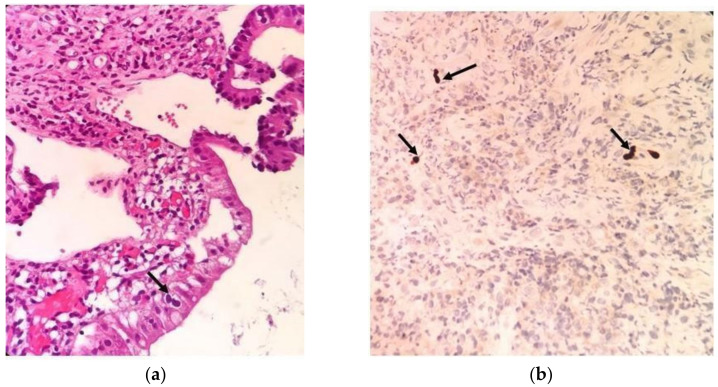
(**a**) Colon surface epithelial cell (arrow) with large basophilic nuclei, large cytoplasmic morphology, prominent nuclear enlargement, and binucleation consistent with the cytopathic effects of CMV (hematoxylin–eosin 200×). (**b**) Specific immunohistochemical (IHC) staining for CMV antigens showing positive cells (arrow) in colorectal tissue (200×) (Bursa Uludag University Faculty of Medicine, Pathology Department).

**Figure 2 viruses-16-00691-f002:**
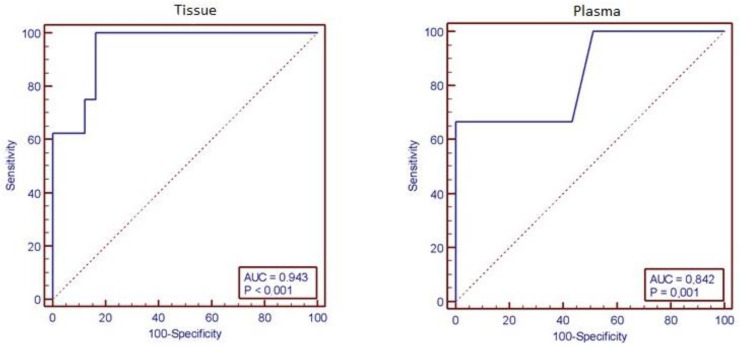
Receiver operating characteristic (ROC) curves were used to evaluate the diagnostic performance of the CMV qPCR method for tissue (area under the curve [AUC] 0.943 [95% CI: 0.87–0.88], *p* < 0.001) and plasma (AUC 0.842 [95% CI: 0.70–0.93]; *p* < 0.001) samples in comparison to immunohistochemistry (IHC).

**Table 1 viruses-16-00691-t001:** Comparison of the presence and levels of CMV DNA in colorectal mucosa tissue based on patient demographic, clinical, and laboratory features.

Variable		Total *n* (%)	CMV DNA		*p*	CMV DNA Load Median (Range) c/mg	*p*
			Detected *n* = 51	Not Detected *n* = 30			
Demographic specifications *n* (%)							
Gender	Male	49 (60.5)	34 (69.4)	15 (30.6)	0.138	20 (0–160,159)	0.533
	Female	32 (39.5)	17 (53.1)	15 (46.9)		20 (0–27,000)	
Age	≤50	48 (59.3)	25 (52.1)	23 (47.9)	0.014	20 (0–27,000)	<0.001
	>50	33 (40.7)	26 (78.8)	7 (21.2)		113 (0–160,159)	
Clinical specifications *n* (%)							
Previous CMV infection	Yes	26 (32.1)	20 (76.9)	6 (23.1)	0.074	77.5 (0–160,159)	0.014
	No	55 (67.9)	31 (56.4)	24 (43.6)		20 (0–27,000)	
GIS symptom	Diarrhea	48 (59.3)	27 (56.3)	21 (43.7)	0.202	0 (0–145,291)	0.228
	Bloody diarrhea	33 (40.7)	24 (72.7)	9 (27.3)		20.0 (0–160,159)	
UC activity	Mild	5 (6.2)	2 (40.0)	3 (60.0)	0.532	0 (0–20)	0.261
	Moderate	20 (24.7)	13 (65.0)	7 (35.0)		45 (0–145,291)	
	Severe	56 (69.1)	36 (64.3)	20 (35.7)		20 (0–160,159)	
Colonoscopic examination *n* (%)							
Ulcer	Yes	70 (86.4)	44 (62.9)	26 (37.1)	0.960	20 (0–160,159)	0.599
	No	11 (13.6)	7 (63.6)	4 (36.4)		20 (0–2460)	
Mucosal inflammation *n* (%)							
Chronic		5 (6.2)	1 (20.0)	4 (80.0)	-	0 (0–20)	0.040
Chronic active		76 (93.8)	50 (65.8)	26 (34.2)		20 (0–160,159)	
Laboratory analysis *n* (%)							
HE	Pos	1 (1.2)	1 (100.0)	0 (0.0)	-	477.0 (-)	-
	Neg	80 (98.8)	50 (62.5)	30 (37.5)		20 (0–160,159)	
IHC	Pos	8 (9.9)	8 (100.0)	0 (0.0)	0.023	25,613.0 (477–160,159)	<0.001
	Neg	73 (90.1)	43 (58.9)	30 (41.1)		20 (0–16,182)	
Plasma CMV DNA^a^ (copies/mL)	Detected	25 (58.1)	22 (88.0)	3 (12.0)	<0.001	500 (0–160,159)	<0.001
	Not detected	18 (41.9)	6 (33.3)	12 (66.7)		0.0 (0–510)	

UC; ulcerative colitis, GIS; gastrointestinal system, Pos; positive, Neg; negative. ^a^ Evaluated in 43 patients. -: The statistical analysis could not be performed.

**Table 2 viruses-16-00691-t002:** Comparison of colorectal mucosa tissue qPCR CMV test results of patients according to the medications used.

Medications	Total *n* = 81 (%)	Tissue CMV DNA		*p*	Tissue CMV DNA Median (Range) Copies/mg	*p*	IHC		*p*
		Detected *n* = 51	Not Detected *n* = 30				Positive *n* = 8	Negative *n* = 73	
Mesalamine	65 (80.2)	40 (61.5)	25 (38.5)	0.806	20 (0–83,160)	0.138	6 (9.2)	59 (90.8)	0.654
Immune-targeting agent	54 (66.7)	37 (68.5)	17 (31.5)	0.222	20 (0–160,159)	0.106	8 (14.8)	46 (85.2)	0.047
Corticosteroid	33 (40.7)	23 (69.7)	10 (30.3)	0.420	26 (0–160,159)	0.059	7 (21.2)	26 (78.8)	0.007
Azathioprine	35 (43.2)	24 (68.6)	11 (31.4)	0.497	20 (0–27,000)	0.798	2 (5.7)	33 (94.3)	0.455
Infliximab or Adalimumab	8 (9.9)	4 (50.0)	4 (50.0)	0.679	10 (0–160,159)	0.671	2 (25.0)	6 (75.0)	0.177

**Table 3 viruses-16-00691-t003:** The test results in all patients and those diagnosed with CMV colitis.

Methods	All Patients *n* = 81 (%)	Patients with CMV Colitis *n* = 21 (%)
	Positive	Positive
H&E	1 (1.2)	1 (4.8)
IHC	8 (9.9)	8 (38.1)
Tissue qPCR	51 (63.0)	21 (100.0)
Plasma qPCR *	25 (58.1)	14 * (93.3)

* Evaluated in 43 patients.

**Table 4 viruses-16-00691-t004:** The test results and outcomes of patients with CMV colitis.

The Course of Patients	*n* = 21 (%)
Positive for H&E, IHC, and qPCR in tissue	8 (38.6)
Positive for only qPCR in tissue	13 (61.9)
Corticosteroid-refractory	13 (61.9)
Pulmonary involvement	2 (9.5)
Clinical remission	19 (90.5)
Colectomy	2 (9.5)
Persistence of plasma CMV DNA	1 (4.8)
Death	1 (4.8)

**Table 5 viruses-16-00691-t005:** Comparison of plasma CMV DNA levels according to tissue qPCR and IHC results.

Tissue	Result	Total *n* = 43 (%)	Plasma CMV DNA Median (Range) Copies/mL	*p*
qPCR CMV DNA	Detected	28 (65.1)	86.5 (0–1937)	<0.001
	Not detected	15 (34.9)	0.0 (0–105)	
IHC	Positive	8 (18.6)	922.0 (20–1937)	0.006
	Negative	35 (81.4)	20.0 (0–578)	

**Table 6 viruses-16-00691-t006:** CMV DNA results were obtained from two tissue samples from 20 patients.

Patient No.	CMV DNA (copies/mg)	
	1. Tissue Sample	2. Tissue Sample
1	477	20
2	16,182	Not detected
3	358	2825
4–20	Not detected	Not detected

## Data Availability

The datasets generated and analyzed during the current study are not publicly available because of privacy policies but are available from the corresponding author upon reasonable request.

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
