# Peer review of "Diagnostic Utility of Cytomegalovirus (CMV) DNA Quantitation in Ulcerative Colitis"

_viruses, 2024, doi:10.3390/v16050691_

Round 1
Reviewer 1 Report
Comments and Suggestions for Authors
The authors present a study seeking to determine whether the need to treat CMV can be predicted in UC patients based on viral load in tissue and/or plasma
The approach is generally sound although for some expts there a many undetected which reduces the statistical power available to the study.
The authors have faithfully reported the data and for the large part I only have minor issues with the study:
1) I cannot determine easily what the whole diagnostic panel for each patient is? i.e. IHC, inclusions, tissue load and plasma load is. It is sort of scattered around the tables but it would be interesting to know if they tally with each other or are they quite disparate. This seems an important question when comparing each approach for utility
2) The biopsy study from table 4 is perhaps the most concerning aspect. It clearly exemplifies (to me) that changing where the biopsy is taken will have a dramatic impact on qPCR data. This is surely a concern that really questions the utility of tissue based qPCR? what do the authors think?
Comments on the Quality of English LanguageEnglish is generally fine but there are instances where very short sentences are added that feel a bit 'note-like'. 'Patients did not have retinitis' for example.
However, overall I did not have major concerns.
Author Response
Reviewer 1

Reviewer 2 Report
Comments and Suggestions for Authors
The authors describe CMV colitis in UC patients. This study is not original and the methodology should be hardly improved.
Lines 83 to 95: First, it is a prospective or a retrospective study? Please clarify. Why not all patients have benefited from a blood sample?
Lines 100-102: I’m not sure that the term “fixed” in 1ml of 0.9% saline is appropriate. If I understand 2 biopsies were sampled for qPCR assay and 2 another for IHC. How do you interpret the results? This should be carefully described.
Lines 117-123: IHC can be interpreted as semiquantitative. Why did you not perform this estimation?
Line 139: Table 1 in the text is to far from the Table in lines 232-234. It is hard to read.
Table 1: Please redo all the calculations presented in Table 1. I'm surprised, for example, that UC activity is not statistically significant between the “CMV detected" versus "CMV not detected" groups. Why do you have chosen the cut-off of 50 for patient’s age? Please justify.
Line 143: I do not understand why CMV DNA load is higher in older patients.
Please detail if you included only previously CMV infected patient or if some patients exhibited primary infection during the study period.
Line 149: are you sure it is insignificant?
Line 158: this sentence is incoherent with the literature.
Line 165: enough effectives for statistical analysis and power?
Line 178: I do not understand.
Line 190-192: the evolution of patients under ganciclovir should be detailed in a table
Line 220-221: what did you decide?
Line 235: Table 2 should be in one piece on the same page
All the discussion should be re-written after correction of the results.
Author Response
Reviewer 2

Round 2
Reviewer 1 Report
Comments and Suggestions for Authors
I would support the inclusion of the additional table and editing to the text
Comments on the Quality of English Languageokay no major problems
Author Response
We would like to express our gratitude to the reviewers for their valuable feedback. The manuscript has been revised to incorporate all of their suggestions. In accordance with the recommendation of reviewer 1, a new table (Table 3) has been added to present test results for all patients, including those diagnosed with CMV colitis.
Minor edits have been made to the English language in the discussion section.
Reviewer 2 Report
Comments and Suggestions for Authors
Dear authors.
I thank you taking time to improve your manuscript. I'm still not convinced by the results you obtained and your results really contradict those of the literature you cite, which leads me either to doubt your methodology, or you need to discuss your results better. And I did not find in the text which result you took when the results of the 2 biopsies were different. IHC should also be semi-quantitative as already reported in the litterature that proposes algorithm to guide therapy.
Soory, I can not accept this work. I let the final decision to the editor.
Author Response
Thank you to the reviewers for their valuable feedback. The manuscript has been revised to incorporate the reviewers' suggestions.
Reviewer 2's comment: I did not find in the text which result you took when the results of the 2 biopsies were different.
Response: If the results of the two biopsy specimens were inconsistent, the positive result was selected.